# Towards Explainable Multimodal Land Cover Segmentation Using Swin Transformer

Islomjon Shukhratov[*1], Mehak Khan[1], and Reza Arghandeh[1]

[1]Western Norway University of Applied Science
{islomjon.shukhratov, mehak.khan, reza.arghandeh}@hvl.no

## Abstract

Recent advancements in Vision Transformers (ViTs) demonstrate strong potential in remote sensing, providing powerful spatial feature representations for complex land cover segmentation tasks. In this study, we explore multimodal data fusion of Synthetic Aperture Radar (SAR) and optical imagery for land cover mapping. We train and evaluate Swin Transformer models and employ explainable AI (xAI) techniques to analyse the contribution of each modality and feature to the model's predictions. We expect to improve the interpretability and robustness of multimodal remote sensing models for land cover segmentation.

## 1 Introduction

Recent advancements in ViTs show state-of-the-art performance in computer vision tasks such as classification and segmentation by leveraging self-attention mechanisms to learn strong representations of spatial features in images [1]. In remote sensing applications, complex land cover patterns and diverse terrain characteristics pose significant challenges, ViTs demonstrate a promise in addressing these challenges, particularly in building segmentation, change detection, scene classification [2, 3].

Multimodal data fusion refers to a combination of several types of data in order to enhance the performance of the deep learning algorithms. In land cover segmentation, the most popular sources for multimodal data fusion are optical and SAR images [4]. However, only a limited number of studies explore the use of transformers for SAR-optical data fusion in land cover classification tasks. In this project, we aim to address this gap by using Sentinel-1 and Sentinel-2 data to train models in one geographic region (the Grand-Est region of France) and evaluate their generalisation in another (Askvoll municipality, Norway). Additionally, we plan to investigate the explainability of the proposed multimodal fusion approach to better understand the model's decision making process.

---

*Corresponding Author.

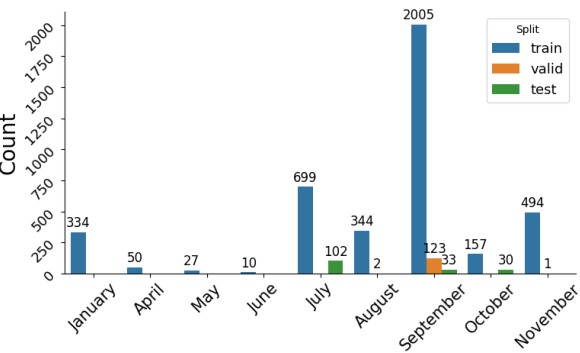

**Figure 1.** Number of samples by month.

## 2 Dataset

We use MultiSenGE dataset provided by Wenger et al. [5]. The dataset covers a large territory in east of the France with the area of 57,433 $km^2$ and consists of Sentinel-1 and Sentinel-2 images for 2020 year, and regional land use and land cover maps provided by OCSGE2-GEOGRANDEST for 2019/2020 years. In total, there are 8157 unique patches of 256x256 pixels generated from 14 tiles of Sentinel-2, and the dataset has 1,012,22 patches for Sentinel-1 and 72,033 patches for Sentinel-2. There are 14 classes in the dataset, but we remap them into 4 main categories to avoid data imbalance issue: urban, agricultural, forest and water areas.

To facilitate the computation resource constraints, we select only the patches that have matching capture date for both sensors, and the resulting number of images is 4411. We split dataset into train, validation and test sets based on tiles, rather than a random split to avoid data leakage and contamination. Consequently, there are 4121, 127 and 167

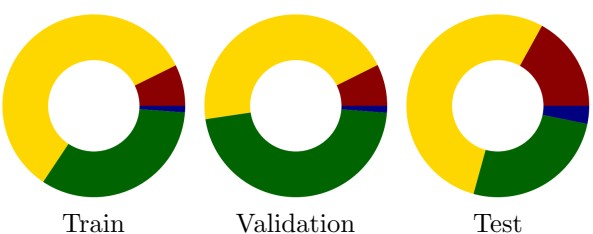

**Figure 2.** Class distribution, red: urban, yellow: agriculture, green: forest, blue: water areas.

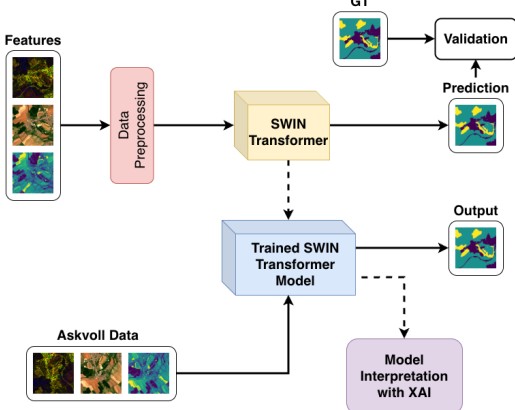

**Figure 3.** Example of the images.

patches for train, validation and test sets. Figure 1 shows monthly distribution of the selected patches.

Additionally, we plan to evaluate the generalization capability of the transformer models on unseen data and different geographic conditions. We designate Askvoll municipality in the Western part of Norway which mainly consists of forests and small lakes in the mountainous terrain. Similar to Multi-SenGE dataset, we collect Sentinel-1 and Sentinel-2 images for the 2022 year, and we use the land cover map provided by Norwegian Institute of Bioeconomy (NIBIO) as a ground truth data.

## 3 Methodology

In our experiments, we intent to use the Shifted Window (Swin) Transformer. Swin Transformer is a hierarchical ViT architecture that computes self-attention within shifted local windows, enabling efficient modelling of both local and global visual dependencies while maintaining linear computational complexity with image size [6]. SWIN transformer is available at small, medium and large sizes, and we train all models for better understanding their capability. Furthermore, to benchmark our results, we aim to train models based on Convolutional Neural Networks (CNN), namely U-Net [7] and DeepLabV3 [8] architectures.

We plan to utilise the VV and VH polarisation bands of the SAR data along with the RGB and NIR bands of the optical imagery, and Normalized Difference Vegetation Index (NDVI) as an additional input feature for training models to perform land cover classification. Figure 3 depicts a sample from the training dataset.

We also plan to investigate the explainability of the SAR–optical multimodal fusion using xAI techniques. Our goal is to understand how each modality and feature contributes to the model's land cover classification decisions, providing insights into the relative importance of SAR and optical inputs and improving the interpretability of multimodal remote sensing models. Figure 4 displays the pipeline of proposed model.

## 4 Expected Results

To evaluate the models' performance, we compute the F1-score and Intersection over Union (IoU) score

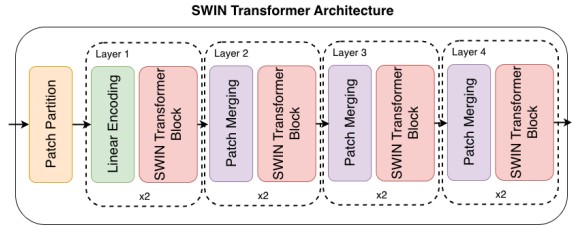

**Figure 4.** Pipeline of the proposed model.

between the predicted image and the ground truth map computed by the following formula:

$$\text{F1 score} = 2 \cdot \frac{\text{Precision} \cdot \text{Recall}}{\text{Precision} + \text{Recall}} \quad (1)$$

$$\text{IoU} = \frac{|\text{Prediction} \cap \text{Ground Truth}|}{|\text{Prediction} \cup \text{Ground Truth}|} \quad (2)$$

We define acceptable performance as a class-average IoU above 70% and F1-score above 80%, We anticipate notable challenges during domain shift evaluation, as the contrasting geographic and environmental conditions. Therefore, we expect 10% drop in metrics compared to the trained dataset.

For xAI interpretability, we apply methods such as Grad-CAM [9], Captum [10] and attention map visualization to assess the spatial relevance and contribution of SAR and optical features to the model's decisions. This is important to understand which modality dominates in a specific class prediction.

## 5 Conclusion

We present an ongoing study on multimodal land cover classification using SAR and optical imagery with vision transformers. The proposed approach demonstrates the potential of multimodal fusion to improve segmentation performance, and future explainability analyses using xAI will provide deeper insights into the contribution of each modality. These results lay the groundwork for developing more interpretable and robust remote sensing models for land cover mapping.

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
