# OpenReview forum: "Towards Explainable Multimodal Land Cover Segmentation Using Swin Transformer"
_NLDL.org/2026/Abstracts_Track — NLDL 2026 Abstracts_

### Official Review · Reviewer_aBgP · 2025-10-31

**Soundness:** 3
**Correctness:** 2
**Rating:** 4
**Confidence:** 4

**Summary:**

The paper proposes to investigate multimodal land segmentation using the Swin Transformer and to quantify how much each modality and attribute contributes to the predictions. The authors plan to use the MultiSenGE dataset and describe their intended model setup and evaluation metrics.

**Strengths:**

•	The abstract provides a clear description of the proposed framework, dataset, and success criteria.

•	The idea and scope are appropriate for the NLDL abstract track.

**Weaknesses:**

•	The naming convention for Swin Transformer should be consistent (Swin, swin, or SWIN?).

•	Wrong submission ID in paper

•	Several abbreviations (e.g., VV, VH) are not defined, which may hinder accessibility for readers outside the immediate subfield.

•	The authors mention that they will use explainable AI (xAI) methods but provide no examples or details of which techniques they plan to apply.

•	The motivation (“why”) could be expressed more clearly. The authors state that they aim to improve interpretability and robustness in multimodal remote sensing, but they do not explain why their proposed framework is particularly well-suited to achieve this.

---

### Official Review · Reviewer_oCoH · 2025-11-02

**Soundness:** 3
**Correctness:** 3
**Rating:** 4
**Confidence:** 4

**Summary:**

This study investigates the use of a contemporary Swin Transformer compared to CNN baselines. This is done on a dataset comprising satellite imagery containing optical imagery (RGB & NIR), and NDVI derivatives of the optical imagery, along with select bands of SAR. Final testing will include domain shifting by evaluating in a geographically distinct area. Accuracy evaluation will rely on F1 and IoU scores. XAI is intended to be used for added explainability regarding the contribution of the various imagery modalities.

**Strengths:**

Overall, the abstract paper is very well formatted, clear and thorough in providing an overview of their research, with no outstanding methodological considerations missing. The core contribution and gap in the literature are clear.

Of importance for deep learning ViT models, the inclusion of data split and imbalance considerations is clearly outlined. The proposed feature maps make sense given the task context.

Evaluation is also clear, with patches seemingly being the atomic unit and evaluation metrics F1 and IoU. The inclusion of XAI is a positive addition, further adding novelty to the study.

**Weaknesses:**

The abstract is largely correct and sound while outlining a very promising study. I only have a few comments:
1) A greater justification for the training, test, and validation split would be ideal. About 93% of the data seems to be used for training, with only 7% remaining for both test and validation. Domain shifting may compensate for the evaluation of generalisability; however, I would recommend that the authors ensure they justify this split in a final paper.

2) To my understanding of the current literature, the use of nomenclature such as "interpretability" (lines: 12, 102, 118) should be limited to intrinsically interpretable models and not the application of ad hoc methods (Rudin, 2019) such as Grad-CAM, Captum, and attention mapping to deep learning models. I would recommend that the authors ensure they use "explainability" when referring to these techniques.


- C. Rudin. “Stop explaining black box machine learning models for high stakes decisions and use interpretable models instead”. In: Nature machine intelligence 1.5 (2019)

---

### Official Review · Reviewer_maPY · 2025-11-03

**Soundness:** 3
**Correctness:** 3
**Rating:** 4
**Confidence:** 4

**Summary:**

This paper investigates multimodal land cover segmentation using Synthetic Aperture Radar (SAR) and optical imagery through the Swin Transformer architecture. The authors aim to fuse Sentinel-1 and Sentinel-2 data for improved accuracy and interpretability in land cover mapping. Their approach involves training Swin Transformers (in multiple model sizes) and benchmarking against CNN-based models such as U-Net and DeepLabV3.

The study uses the MultiSenGE dataset (Grand-Est, France) and tests generalization on Askvoll, Norway, addressing domain shift challenges. Additionally, explainable AI (xAI) techniques (Grad-CAM, Captum, attention maps) are employed to interpret the contributions of SAR and optical features to segmentation performance. The authors define success as achieving class-mean IoU > 70% and F1 > 80%. The expected outcome is an interpretable and robust multimodal Swin Transformer model for land cover mapping.

**Strengths:**

1. Incorporates complementary sensing modalities that enhance robustness under diverse environmental conditions (e.g., cloud cover).

2. Combines deep learning performance with interpretability, using established xAI tools (Grad-CAM, Captum).

3. Evaluates models on geographically distinct datasets (France vs. Norway), showing awareness of domain shift issues and real-world deployment challenges.

4. Tile-based splitting avoids data leakage, and label remapping mitigates class imbalance.

**Weaknesses:**

1. The abstract outlines the design and expectations but does not provide actual performance outcomes, making it hard to assess effectiveness or novelty empirically.

2. Uses established models (Swin Transformer, standard CNNs) and standard fusion and xAI techniques; innovation lies mainly in application rather than architecture or algorithmic design.

3. Dataset largely dominated by certain classes (e.g., agriculture/forest), which could bias performance; this is not extensively discussed.

4. The pipeline does not clearly detail how SAR and optical modalities are fused (early, mid, or late fusion), which is crucial for interpretability and performance understanding.

5. While Grad-CAM and Captum are valuable, deeper modality attribution or counterfactual analyses could provide richer insight into model decision-making.

---

### Decision · Program_Chairs · 2025-11-05

**Decision:**

Accept

**Comment:**

The abstract is of interest to the community and should be presented at the conference.